# A Smoothed Analysis of the Greedy Algorithm for the Linear Contextual Bandit Problem

**Sampath Kannan**
University of Pennsylvania

**Jamie Morgenstern**
Georgia Tech

**Aaron Roth**
University of Pennsylvania

**Bo Waggoner**
Microsoft Research, NYC

**Zhiwei Steven Wu**
University of Minnesota

## Abstract

Bandit learning is characterized by the tension between long-term exploration and short-term exploitation. However, as has recently been noted, in settings in which the choices of the learning algorithm correspond to important decisions about individual people (such as criminal recidivism prediction, lending, and sequential drug trials), exploration corresponds to explicitly sacrificing the well-being of one individual for the potential future benefit of others. In such settings, one might like to run a "greedy" algorithm, which always makes the optimal decision for the individuals at hand — but doing this can result in a catastrophic failure to learn. In this paper, we consider the linear contextual bandit problem and revisit the performance of the greedy algorithm. We give a smoothed analysis, showing that even when contexts may be chosen by an adversary, small perturbations of the adversary's choices suffice for the algorithm to achieve "no regret", perhaps (depending on the specifics of the setting) with a constant amount of initial training data. This suggests that in slightly perturbed environments, exploration and exploitation need not be in conflict in the linear setting.[1]

## 1 Introduction

Learning algorithms often need to operate in partial feedback settings (also known as *bandit* settings), in which the decisions of the algorithm determine the data that it observes. Many real-world application domains of machine learning have this flavor. Predictive policing algorithms [Rudin, 2013] deploy police officers and receive feedback about crimes committed and observed in areas the algorithm chose to deploy officers. Lending algorithms [Byrnes, 2016] observe whether individuals who were granted loans pay them back, but do not get to observe counterfactuals: would an individual not granted a loan have repaid such a loan? Algorithms which inform bail and parole decisions [Barry-Jester et al., 2015] observe whether individuals who are released go on to recidivate, but do not get to observe whether individuals who remain incarcerated *would* have committed crimes had they been released. Algorithms assigning drugs to patients in clinical trials do not get to observe the effects of the drugs that were not assigned to particular patients.

Learning in partial feedback settings faces the well-understood tension between *exploration* and *exploitation*. In order to perform well, the algorithms need at some point to exploit the information they have gathered and make the best decisions they can. But they also need to explore: to make decisions which do not seem optimal according to the algorithm's current point-predictions, in order to gather more information about less explored portions of the decision space.

However, in practice, decision systems often do not explicitly explore, for a number of reasons. Exploration is important for maximizing long-run performance, but decision makers might be myopic — more interested in their short-term reward. In other situations, the decisions made at each round affect the lives of individuals, and explicit exploration might be objectionable on its face: it can be considered immoral to harm an individual today (explicitly sacrificing present utility) for a potential benefit to future individuals (long-term learning rates) [Bird et al., 2016, Bastani et al., 2017]. For example, in a medical trial, it may be repugnant to knowingly assign a patient a drug that is thought to be sub-optimal (or even dangerous) given the current state of knowledge, simply to increase statistical certainty. In a parole scenario, we may not want to release a criminal that we estimate is at high risk for committing violent crime.

On the other hand, a lack of exploration can lead to a catastrophic failure to learn, which is highly undesirable – and which can also lead to unfairness. A lack of exploration (and a corresponding failure to correctly learn about crime statistics) has been blamed as a source of "unfairness" in predictive policing algorithms [Ensign et al., 2017]. In this paper, we seek to quantify how costly we should expect a lack of exploration to be when the instances are not entirely worst-case. In other words: is myopia a friction that we should generically expect to quickly be overcome, or is it really a long-term obstacle to learning? Empirical evaluation shows that greedy algorithms often do well — even outperforming algorithms with explicit exploration [Bietti et al., 2018]. Our work provides a theoretical explanation for this phenomenon.

## 1.1 Our Results

We study the *linear contextual bandits problem*, which informally, represents the following learning scenario which takes place over a sequence of rounds $t$ (formal definitions appear in Section 2). At each round $t$, the learner must make a decision amongst $k$ choices, which are represented by *contexts* $x_i^t \in \mathbb{R}^d$. If the learner chooses action $i_t$ at round $t$, he observes a reward $r_{i_t}^t$ — but does not observe the rewards corresponding to choices not taken. The rewards are stochastic, and their expectations are governed by unknown linear functions of the contexts. For an unknown set of parameters $\beta_i \in \mathbb{R}^d$, $\mathbb{E}[r_i^t] = \beta_i \cdot x_i^t$. We consider two variants of the problem: in one (the *single parameter setting*), all of the rewards are governed by the *same* linear function: $\beta_1 = \ldots = \beta_k = \beta$. In the other (the *multiple parameter setting*), the parameter vectors $\beta_i$ for each choice can be distinct. Normally, these two settings are equivalent to one another (up to a factor of $k$ in the problem dimension) — but as we show, in our case, they have distinct properties[2]. The single-parameter setting can model, for example, the choice of which of some subset of individuals should participate in a particular clinical trial. The multi-parameter setting can model, for example, the risk of criminal recidivism amongst different individuals who come from different backgrounds, when observable features correlate differently to crime risk amongst different groups of individuals.

We study the *greedy algorithm*, which trains least-squares estimates $\hat{\beta}_i^t$ on the current set of observations, and at each round, picks the arm with the highest predicted reward: $i_t = \arg\max_i \hat{\beta}_i^t \cdot x_i^t$. In the single parameter setting, greedy simply maintains a single estimate $\hat{\beta}^t$.

It is well known that the greedy algorithm does not obtain any non-trivial worst-case regret bound. We give a smoothed analysis which shows that the worst case is brittle, however. Specifically, we consider a model in which the contexts $x_i^t$ are chosen at each round by an adaptive adversary, but are then perturbed by independent Gaussian perturbations in each coordinate, with standard deviation $\sigma$. We show that under smoothed analysis, there is a qualitative distinction between the single parameter and multiple parameter settings:

1. In the single parameter setting (Section 4), the greedy algorithm with high probability obtains regret bounded by $\tilde{O}\left(\frac{\sqrt{T}d}{\sigma^2}\right)$ over $T$ rounds.

2. In the multiple parameter setting (Section 5), the greedy algorithm requires a "warm start" – that is, to start with a small number of observations for each action – to obtain non-trivial regret bounds, even when facing a perturbed adversary. We show that if the warm start provides for each arm a small number of examples (depending polynomially on fixed parameters of the instance, like $1/\sigma$, $d$, $k$, and $1/(\min_i ||\beta_i||)$), that may themselves be

chosen by an adversary and perturbed, then with high probability greedy obtains regret $\tilde{O}\left(\frac{\sqrt{Tk}}{\sigma^2}\right)$. Moreover, this warm start is necessary: we give lower bounds showing that if the greedy algorithm is not initialized with a number of examples $n$ that grows polynomially with both $1/\sigma$ and with $1/\min_i ||\beta_i||$, then there are simple fixed instances that force the algorithm to have regret growing linearly with $T$, with constant probability. (See Section 6 for a formal statement of the lower bounds.)

Our results extend beyond this particular perturbed adversary: we give general conditions on the distribution over contexts which imply our regret bounds. All missing proofs can be found in the full version.

## 1.2 Related Work

The most closely related piece of work (from which we take direct inspiration) is Bastani et al. [2017], who, in a stochastic setting, give conditions on the sampling distribution over contexts that causes the greedy algorithm to have diminishing regret in a closely related but incomparable version of the two-armed linear contextual bandits problem[3]. The conditions on the context distribution given in that work are restrictive, however. They imply, for example, that every linear policy (and in particular the optimal policy) will choose each action with constant probability bounded away from zero. When translated to our perturbed adversarial setting, the distributional conditions of Bastani et al. [2017] do not imply regret bounds that are sub-exponential in either the perturbation magnitude $\sigma$ or the dimension $d$ of the problem. There is also strong empirical evidence that exploration free algorithms perform well on real datasets: [Bietti et al., 2018]. Our work can be viewed as providing an explanation of this phenomenon. Finally, building on our work, [Raghavan et al., 2018] use the same diversity condition that we introduce in this paper to show a stronger result in a more restrictive setting. They show that in the single parameter setting, when one further assumes that 1) the linear parameter is drawn from a Bayesian prior that is not too concentrated, 2) the contexts are drawn i.i.d. from a fixed distribution and then perturbed, and 3) that the algorithm is allowed to make its decisions in "batches" of polylog$(d, t)/\sigma^2$ many rounds, then the greedy algorithm is essentially *instance optimal* in terms of Bayesian regret, and moreover, that its regret grows at a rate of $O(T^{1/3})$ in the worst case. In contrast, we make substantially weaker assumptions (the parameter vector and contexts can be worst case, we need not be in the single parameter setting, and we don't need batches), but prove a worse regret bound of $O(T^{1/2})$, without a guarantee of instance optimality.

A large literature focuses on designing no-regret algorithms for contextual bandit problems (e.g. Li et al. [2010], Agarwal et al. [2014], Li et al. [2011]), particularly for linear contextual bandits (e.g. [Chu et al., 2011, Abbasi-Yadkori et al., 2011]). Some of these (e.g. Syrgkanis et al. [2016]) use "follow the perturbed leader" style algorithms, which invite a natural comparison to our setting. However, the phenomenon we are exploiting is quite different. It is very important in our setting that the perturbations are added by nature, and if the perturbations were instead added by our algorithm (against worst-case contexts), the regret guarantee would cease to hold. To see this, note that against worst-case adversaries, the single parameter and multiple parameter settings are equivalent to one another — but in our smoothed setting, we prove a qualitative separation.

We defer further related work, including work on smoothed analysis and algorithmic fairness, to the full version.

## 2 Model and Preliminaries

We now introduce the notation and definitions we use for this work. For a vector $x$, $\|x\|$ represents its Euclidean norm. We consider two variants of the $k$-arm linear contextual bandits problem. The first setting has a single $d$-dimensional parameter vector $\beta$ which governs rewards for all contexts $x \in \mathbb{R}^d$; the second has $k$ distinct parameter vectors $\beta_i \in \mathbb{R}^d$ governing the rewards for different arms.

In rounds $t$, contexts $x_1^t, \ldots, x_k^t$, are presented, where $x_i^t \in \mathbb{R}^d$ is treated as a row vector unless otherwise noted. The learner chooses an arm $i^t \in \{1, \ldots, k\}$, and obtains $s^2$-subgaussian[4] reward $r^t$ whose mean satisfies $\mathbb{E}[r^t] = \beta \cdot x_{i^t}^t$ in the single parameter setting and $\mathbb{E}[r^t] = \beta_{i^t} \cdot x_{i^t}^t$ in the multi-parameter setting. The regret of a sequence of actions and contexts of length $T$ is (again, in the single parameter setting all $\beta_i = \beta$):

$$\text{Regret} = \text{Regret}(x^1, i^1, \ldots, x^T, i^T) = \sum_{t=1}^{T} \left( \max_i \beta_i \cdot x_i^t - \beta_{i^t} \cdot x_{i^t}^t \right).$$

We next formalize the history or transcript of an algorithm on a sequence of contexts. A *history entry* is a member of $\mathcal{H} = \left( \mathbb{R}^d \right)^k \times \{1, \ldots, k\} \times \mathbb{R}$. A *history* is a list of history entries, i.e. a member of $\mathcal{H}^*$. Given a history $H \in \mathcal{H}^T$, entry $t$ is denoted $h^t = (x_1, \ldots, x_k, i^t, r_{i^t}^t)$.

Formally, an adaptive *adversary* $\mathcal{A}$ is a (possibly randomized) algorithm that maps a history to $k$ contexts: $\mathcal{A} : \mathcal{H}^* \to \left( \mathbb{R}^d \right)^k$. We denote the output of the adversary by $(\mu_1, \mu_2, \ldots, \mu_k)$[5] We assume that $\|\mu_i\| \leq 1$ always. Next we define the notion of a perturbed adversary, which encompasses both stages of the context-generation process.

**Definition 1** (Perturbed Adversary). For any adversary $\mathcal{A}$, the $\sigma$-*perturbed adversary* $\mathcal{A}_\sigma$ is defined by the following process. In round $t$:

1. Given history $H^{t-1} \in \mathcal{H}^{t-1}$, let $\mu_1^t, \ldots, \mu_k^t = \mathcal{A}(H^{t-1})$.
2. Perturbations $e_1^t, \ldots, e_k^t$ are drawn independently from $\mathcal{N}(0, \sigma^2 I)$.
3. Output the list of contexts $(x_1^t, \ldots, x_k^t) = (\mu_1^t + e_1^t, \ldots, \mu_k^t + e_k^t)$.

We define a perturbed adversary to be *R-bounded* if with probability 1, $\|x_i^t\| \leq R$ for all $i$ and $t$ and all histories. We call perturbations $(r, \delta)$-*centrally bounded* if, for each history, and fixed unit vectors $w_1, \ldots, w_k$ (possibly all equal), we have with probability $1 - \delta$ that $\max_{i=1,\ldots,k} w_i \cdot e_i^t \leq r$.

We can interpret the output of a perturbed adversary as being a mild perturbation of the (unperturbed) adaptive adversary when the magnitude of the perturbations is smaller than the magnitude of the original context choices $\mu_i$ themselves. Said another way, we can think of the perturbations as being mild when they do not substantially increase the norms of the contexts with probability at least $1 - \delta$. This will be the case throughout the run of the algorithm (via a union bound over $T$) when $\sigma \leq \tilde{O}(1/\sqrt{d})$. We refer to this case as the "low perturbation regime". We view it as the most interesting case because otherwise, the perturbations tend to be large enough to overwhelm the adversarial choices and the problem becomes easier. Here we focus on presenting results for the low perturbation regime, leaving the rest to the full version.

## 3 Proof Approach and Key Conditions

Our goal will be to show that the greedy algorithm achieves no regret against any perturbed adversary in both the single-parameter and multiple-parameter settings. The key idea is to show that the distribution on contexts generated by perturbed adversaries satisfy certain conditions which suffice to prove a regret bound. The conditions we work with are related to (but substantially weaker than) the conditions shown to be sufficient for a no regret guarantee by Bastani et al. [2017].

The first key condition, *diversity* of contexts, considers the positive semidefinite $d \times d$ matrix $\mathbb{E}[x^\mathsf{T} x]$ for a context $x$, and asks for a lower bound on its minimum eigenvalue. This implies the distribution over $x$ has non-trivial *variance* in all directions, which is necessary for the least squares estimator to converge to the underlying parameter $\beta$. It implies that observations of $\beta \cdot x$ convey information about $\beta$ in all directions.

However, we only observe the rewards for contexts $x$ conditioned on Greedy selecting them: we see a biased (*conditional*) distribution on $x$. Thus we need the diversity condition to hold on these conditional distributions.

**Condition 1** (Diversity). *Let $e \sim \mathcal{D}$ on $\mathbb{R}^d$ and let $r, \lambda_0 \geq 0$. We call $\mathcal{D}$ $(r, \lambda_0)$-diverse if for all $\hat{\beta}$, all $\mu$ with $\|\mu\| \leq 1$, and all $\hat{b} \leq r\|\hat{\beta}\|$, for $x = \mu + e$:*

$$\lambda_{min}\left(\mathop{\mathbb{E}}_{e \sim \mathcal{D}}\left[x^{\mathsf{T}}x \mid \hat{\beta} \cdot e \geq \hat{b}\right]\right) \geq \lambda_0.$$

In the single parameter setting, diversity will imply a regret guarantee: when *any* arm is pulled, the context-reward pair gives useful information about all components of the (single) parameter $\beta$. In the multiple parameter setting, diversity will suffice to guarantee that the learner's estimate of arm $i$'s parameter vector converges to $\beta_i$ as a function of the number of times arm $i$ is pulled; but alone it does not cause arm $i$ to be pulled sufficiently often (even in rounds where $i$ is the best alternative, when failing to pull it will cause our algorithm to suffer regret).

Thus the multiple parameter setting will require a second key condition, *margins*. Margins will imply that conditioned on an arm being optimal on a given round, there is a non-trivial probability (over the randomness in the contexts) that Greedy perceives it to be optimal based on current estimates $\{\hat{\beta}_i^t\}$, so long as the current estimates achieve at least some constant baseline accuracy. A small initial training set can guarantee that initial estimates achieve constant error, and so the margin condition implies that Greedy will continue to explore arms with a frequency that is proportional to the number of rounds for which they are optimal; then diversity implies that estimates of those arms' parameters will improve quickly (without promising anything about arms that are rarely optimal – and hence inconsequential for regret).

**Condition 2** (Conditional Margins)**.** *Let $e \sim \mathcal{D}$ and let $r, \alpha, \gamma \geq 0$. We say $\mathcal{D}$ has $(r, \alpha, \gamma)$ margins if for all $\beta \neq 0$ and $b \leq r\|\beta\|$,*

$$\mathbb{P}\left[\beta \cdot e > b + \alpha\|\beta\| \mid \beta \cdot e \geq b\right] \geq \gamma.$$

So, on rounds for which arm $i$ has largest expected reward, with probability at least $\gamma$ its expected reward is largest by at least some margin ($\alpha\|\beta\|$). If Greedy has sufficiently accurate estimates $\{\hat{\beta}_i^t\}$, this implies that Greedy will pull arm $i$. We say a perturbed adversary satisfies the diversity and margin conditions if the distributions of $e_i^t$ are independent and satisfy these conditions for all $i, t$.

We will show the diversity condition implies no-regret in single-parameter settings, and the diversity and margin conditions imply no-regret in multi-parameter settings. We further show that the perturbation distribution $\mathcal{N}(0, \sigma^2 I)$ satisfies these conditions. We note that our choice of Gaussian perturbations was convenient and natural but not necessary (other perturbation distributions also satisfy our conditions, implying similar results for those perturbations).

**Complications: extreme perturbation realizations.** When the realizations of the Gaussian perturbations have extremely large magnitude, the diversity and margin conditions will not hold[6]. This is potentially problematic, because the probabilistic conditioning in both conditions increases the likelihood that the perturbations will be large. This is the role of the parameter $r$ in both conditions: to provide a reasonable upper bound on the threshold that a perturbation variable should not exceed. exceed. In the succeeding sections, we will use conditions we call "good" to formalize the intuition that this is unlikely to happen often, when the perturbations satisfy a centrally-bounded condition.

## 4 Single Parameter Setting

We define the "Greedy Algorithm" as the algorithm which myopically pulls the "best" arm at each round according to the predictions of the classic least-squares estimator. Let $X^t$ denote the $(t-1) \times d$ design matrix at time $t$, in which each row $t'$ is some observed context $x_{i^{t'}}^{t'}$ where arm $i^{t'}$ was selected at round $t' < t$. The corresponding vector of rewards is denoted $y^t = (r_{i^1}^1, \ldots, r_{i^{t-1}}^{t-1})$. The transposes of a matrix $Z$ and vector $z$ are denoted $Z^{\mathsf{T}}$ and $z^{\mathsf{T}}$. At each round $t$, Greedy first computes the least-squares estimator based on the historical contexts and rewards: $\hat{\beta}^t \in \arg\min_\beta \|X^t\beta - y^t\|_2^2$, and then greedily selects the arm with the highest estimated reward: $i^t = \arg\max_i \hat{\beta}^t \cdot x_i^t$. We defer the formal description of the algorithm to the full version.

**"Reasonable" rounds.** As discussed in Section 2, the diversity condition will only hold when an arm's perturbations $e_i^t$ are not too large; we formalize these "good" situations here. Fix a round $t$,

the current Greedy hypothesis $\hat{\beta}^t$, and any choices of the adversary $\mu_1^t, \ldots, \mu_k^t$ conditioned on the entire history up to round $t$. Now each value $\hat{\beta}^t x_i^t = \hat{\beta}^t \mu_i^t + \hat{\beta}^t e_i^t$ is a random variable, and Greedy selects the arm corresponding to the largest realized value. In particular, we define the "threshold" for Greedy to pull $i$ as follows.

**Definition 2.** Fix a round $t$, Greedy's hypothesis $\hat{\beta}^t$, and the adversary's choices $\mu_1^t, \ldots, \mu_k^t$. We define $\hat{c}_i^t := \max_{j \neq i} \hat{\beta}^t \cdot x_j^t$. We say a realization of $\hat{c}_i^t$ is $r$-$\widehat{good}$ if $\hat{c}_i^t \leq \hat{\beta}^t \cdot \mu_i^t + r\|\hat{\beta}^t\|$.

The "hat" on $\widehat{good}$ corresponds to those on $\hat{c}_i^t$ and $\hat{\beta}^t$. In the multiple parameter setting we will use analogous conditions without the hats. Notice that $\hat{c}_i^t$ is a random variable that depends on all the perturbations $e_j^t$ for $j \neq i$, and Greedy pulls $i$ if and only if $\hat{\beta}^t x_i^t \geq \hat{c}_i^t$. The event that $\hat{c}_i^t$ is $r$-good is determined by the perturbations $e_{i'}^t$ of all arms $i' \neq i$. Intuitively, if $\hat{c}_i^t$ is $r$-$\widehat{good}$, then $e_i^t$ need not be too large for arm $i$ to be selected.

## 4.1 Regret framework for perturbed adversaries

We first observe an upper-bound on Greedy's regret as a function of the distance between $\hat{\beta}^t$ and the true model $\beta$. This allows us to focus on the diversity condition, which will guarantee that this distance shrinks. Let $i^*(t) = \arg\max_i \beta \cdot x_i^t$, the optimal arm at time $t$.

**Lemma 4.1.** *Suppose for all $i, t$ that $\|x_i^t\| \leq R$. In the single-parameter setting, for any $t_{\min} \in [T]$, we have:*
$$Regret(x^1, i^1, \ldots, x^T, i^T) \leq 2Rt_{\min} + 2R \sum_{t=t_{\min}}^{T} \left\| \beta - \hat{\beta}^t \right\|.$$

To apply Lemma 4.1, we need to show that estimates $\hat{\beta}^t \to \beta$ quickly. The key idea is that if the input contexts are "diverse" enough (captured formally by Definition 1), we will be able to infer $\beta$. Lemma 4.2 shows $\hat{\beta}^t$ approaches $\beta$ at a rate governed by the minimum eigenvalue of the design matrix.

**Lemma 4.2.** *Fix a round $t$ and let $Z^t = (X^t)^\mathsf{T} X^t$. Suppose all contexts satisfy $\|x_i^t\| \leq R$ and recall that rewards are $s^2$-subgaussian. Then with probability $1 - \delta$ over the randomness in rewards, we have*
$$\|\beta - \hat{\beta}^t\| \leq \frac{\sqrt{2tdRs^2 \ln(td/\delta)}}{\lambda_{min}(Z^t)}.$$

Observe that the matrix $Z^t = \sum_{t' \leq t} (x_i^{t'})^\mathsf{T} x_i^{t'}$. The next step is to show that $\lambda_{\min}(Z^t)$ grows at a rate of $\Theta(t)$ with high probability, which will imply via Lemma 4.2 that $\|\beta - \hat{\beta}^t\| \leq O(1/\sqrt{t})$, fixing all other parameters. This is proven in the following key result, Lemma 4.3. The proof uses a concentration result for the minimum eigenvalue to show that $\lambda_{\min}(Z^t)$ grows at a rate $\Theta(t)$ with high probability. This relies crucially on the $(r, \lambda_0)$ diversity condition, which intuitively lower-bounds the expected increase in $\lambda_{\min}(Z^t)$ at each round. The details are more complicated, as this increase only holds when Greedy's choice of $i$ has an $r$-$\widehat{good}$ $\hat{c}_i^t$; we show this happens with constant probability for an $(r, 1/2)$-centrally bounded adversary.

**Lemma 4.3.** *For Greedy in the single parameter setting with an $R$-bounded, $(r, 1/2)$-centrally bounded, $(r, \lambda_0)$-diverse adversary, we have with probability $1 - \delta$ that for all $t \geq \max\{0, \frac{20R^2}{\lambda_0} \ln(\frac{20R^2}{\lambda_0 d\delta})\}$, we have $\lambda_{min}(Z^t) \geq \frac{t\lambda_0}{4}$.*

Combining these results gives a bound on the regret of Greedy against general perturbed adversaries.

**The Gaussian, $\sigma$-perturbed adversary.** We need to show that our $\sigma$-perturbed adversary satisfies the diversity condition (and another technical condition that we defer to the supplementary materials). For the diversity condition, we show that the diversity parameter $\lambda$ can be lower bounded by the variance of a single-dimensional truncated Gaussian, then analyze this variance using tight Gaussian tail bounds. Our proof makes use of the careful choice of truncations of $\mathcal{A}_\sigma'$ using a different orthonormal change of basis each round, which maintains the perturbation's Gaussian distribution but allows the form of the conditioning to be much simplified. Finally, we arrive at the main result for this section:

**Theorem 4.1.** *In the single parameter setting against the $\sigma$-perturbed adversary $\mathcal{A}_\sigma$, fix any choice of parameters such that $\sigma \leq \frac{1}{2\sqrt{2d\ln(Tkd/\delta)}}$ (the low perturbation regime) and $d \leq e^{O(s^2 T)}$. With probability at least $1 - \delta$, Greedy has*

$$Regret \leq O\left(\frac{\sqrt{Tds^2 \ln(Td/\delta)}\ln(k)}{\sigma^2}\right)$$

*where $d$ is the dimension of contexts, $k$ is the number of arms, rewards are $s^2$-subgaussian, and in all cases $O(\cdot)$ hides an absolute constant.*

## 5 Multiple Parameter Setting

In the multi-parameter setting, we cannot hope for the greedy algorithm to achieve vanishing regret without any initial information, as it never learns about parameters of arms it does not pull (formalized in a lower bound in Section 6). If, however, Greedy receives a small amount of initial information in the form of a constant number of $n$ samples $(x_i, r_i)$ for each arm $i$, perturbations will imply vanishing regret. We refer to this as an $n$-sample "warm start" to Greedy. (See the full version for a formal description of the algorithm.)

For this setting, we show that the diversity and margin conditions together on a generic bounded adversary imply low regret. We then leverage this to give regret bounds for the Gaussian adversary $\mathcal{A}_\sigma$. As discussed in Section 3, the key idea is as follows. Analogous to the single parameter setting, the diversity condition implies that additional datapoints we collect *for an arm* improve the accuracy of its estimate $\hat{\beta}_i^t$. Meanwhile, the *margin condition* implies that for sufficiently accurate estimates, when an arm is optimal ($\beta_i x_i^t$ is largest), the perturbations have a good chance of causing Greedy to pull that arm ($\hat{\beta}_i^t x_i^t$ is largest). Thus, the initial data sample kickstarts Greedy with reasonably accurate estimates, causing it to regularly pull optimal arms and accrue more data points, thus becoming more accurate.

**Notation and preliminaries.** Recall that $i^t$ is the arm pulled by Greedy at round $t$, i.e. $i^t = \arg\max_i \hat{\beta}_i^t \cdot x_i^t$. Similarly let $i^*(t)$ be the optimal arm at round $t$, i.e. $i^*(t) = \arg\max_i \beta_i \cdot x_i^t$. Let $n_i(t)$ be the number of times arm $i$ is pulled prior to round $t$, including the warm start (so $n_i(1)$ will be nonzero). Let $S_i = \{t : i^t = i\}$ and let $S_i^* = \{t : i^*(t) = i\}$, the rounds where $i$ is pulled and is optimal respectively.

Recall that $\hat{c}_i^t$ is a threshold that $i$ must exceed to be pulled by Greedy, and the $r\text{-}\widehat{good}$ condition captures cases where this can happen without the perturbation $e_i^t$ being too large. We now define this condition formally for the multiple parameter case. We also need a similar threshold $c_i^t$ that $i$ must exceed to be the *optimal* arm, and an analogous $r$-good condition.

**Definition 3.** Fix a round $t$, the current Greedy hypotheses $\hat{\beta}_1^t, \ldots, \hat{\beta}_k^t$, and choices of an adversary $\mu_1^t, \ldots, \mu_k^t$. Define $\hat{c}_i^t := \max_{j\neq i} \hat{\beta}_j^t \cdot x_j^t$, a random variable depending on $\{e_j^t : j \neq i\}$. Say an outcome of $\hat{c}_i^t$ is $r\text{-}\widehat{good}$ if $\hat{c}_i^t \leq \hat{\beta}_i^t \cdot \mu_i^t + r\|\hat{\beta}_i^t\|$. Similarly, define $c_i^t := \max_{j\neq i} \beta_j \cdot x_j^t$ and say an outcome of $c_i^t$ is $r$-good if $c_i^t \leq \beta_i \cdot \mu_i^t + r\|\beta_i\|$.

### 5.1 Regret framework for perturbed adversaries

Similarly to Lemma 4.1, here the regret of Greedy shrinks as each $\hat{\beta}_i^t \to \beta_i$. The proof is essentially identical, but in this case, we prove this for each arm $i \in [k]$.

**Lemma 5.1.** *In the multiple parameter setting, the regret of Greedy is bounded by $\sum_{i=1}^k Regret_i(T)$ with*

$$Regret_i(T) = R\left(\sum_{t \in S_i}\left\|\beta_i - \hat{\beta}_i^t\right\|\right) + R\left(\sum_{t \in S_i^*}\left\|\beta_i - \hat{\beta}_i^t\right\|\right).$$

As in the single parameter setting, the diversity condition implies that with enough observations $n_i(t)$ for arm $i$, we have $\|\beta_i - \hat{\beta}_i^t\| \leq O(\frac{1}{\sqrt{n_i(t)}})$. We omit the details as they are analogous to the single parameter case, and move on to the margin condition.

We wish to capture the benefits of the margin condition, i.e. that arms which are often optimal are also actually pulled often by Greedy. The first step is to leverage the margin condition to argue that when arm $i$ is optimal (and $c_i^t$ is $r$-good), it is optimal by a significant margin ($\alpha\|\beta_i\|$) with a significant probability ($\gamma$). Combining this with accurate initial estimates implies that it will actually be pulled by Greedy.

**Lemma 5.2.** *Suppose the perturbed adversary is $R$-bounded and has $(r, \alpha, \gamma)$ margins for some $r \leq R$. Consider any round $t$ where for all $j$ we have $\|\beta_j - \hat{\beta}_j^t\| \leq \frac{\alpha \min_{j'} \|\beta_{j'}\|}{2R}$. Then*

$$\mathbb{P}\left[i^t = i \mid i^*(t) = i, c_i^t \text{ is } r\text{-good}\right] \geq \gamma.$$

Recall that $S_i$, $S_i^*$ are respectively the set of rounds in which $i^t = i$ (Greedy pulls arm $i$) and $i^*(t) = i$ (arm $i$ is optimal), respectively. The following key result leverages the margin condition to argue that, if $i$ is optimal for a significant number of rounds, then it is pulled by Greedy. This is vital to a good regret bound because it shows that $n_i(t)$, the number of samples from arm $i$, is steadily increasing in $t$ if $i$ is often optimal, which we know from the diversity condition implies that the estimate $\hat{\beta}_i^t$ is converging.

**Lemma 5.3.** *Consider an $R$-bounded perturbed adversary with $(r, \alpha, \gamma)$ margins and assume $\|\beta_i - \hat{\beta}_i^t\| \leq \frac{\alpha \min_j \|\beta_j\|}{2R}$ for all $i$ and $t$. With probability at least $1 - \delta$, we have for all natural numbers $N$, $|\{t \in S_i^* : n_i(t) = N\}| \leq \frac{5}{\gamma} \ln \frac{2}{\delta}$. That is, arm $i$ can be optimal at most $\frac{5}{\gamma} \ln \frac{2}{\delta}$ times before being pulled by Greedy.*

**The Gaussian, $\sigma$-perturbed adversary**    At a high level, all that remains to complete our analysis is to prove that our perturbed adversary $\mathcal{A}_\sigma$ produces distributions that satisfy our margin condition. There are some complications that make the details of this argument slightly circuitous, that we defer to the full version in the supplement (we first prove this result for an adversary that uses a truncated Gaussian distribution, and hence always produces bounded contexts, and then use this to argue that our actual adversary also has the properties that we need). Once this is proven, we obtain the main result of the multiple parameter setting. In particular, in the small-perturbation regime, a constant-size warm start (i.e. independent of $T$, as long as $\sigma$ is small) suffices to initialize Greedy such that, with high probability, it can obtain $\tilde{O}(\sqrt{T})$ regret.

**Theorem 5.1.** *In the multiple parameter setting, against the $\sigma$-perturbed adversary $\mathcal{A}_\sigma$, with a warm start of size*

$$n \geq \Omega\left(\frac{ds^2 \ln(\frac{dks^2}{\delta\sigma \min_j \|\beta_j\|^2})}{\sigma^{12} \min_j \|\beta_j\|^2}\right),$$

*for any setting of parameters such that $\sigma \leq \frac{1}{3\sqrt{d \ln(2Tkd/\delta)}}$, Greedy satisfies with probability $1 - \delta$*

$$Regret \leq O\left(\frac{\sqrt{Tkds^2}(\ln \frac{Tkd}{\delta})^{3/2}}{\sigma^2}\right)$$

*where $d$ is the dimension of the contexts, $k$ is the number of arms, rewards are $s^2$-subgaussian, and in all cases $O(\cdot)$, $\Omega(\cdot)$ hide absolute constants.*

# 6    Lower Bounds for the Multi-Parameter Setting

Finally, in this section, we show that our results for the multi-parameter setting are qualitatively tight. Namely, Greedy can be forced to suffer linear regret in the multi-parameter setting unless it is given a "warm start" that scales polynomially with $\frac{1}{\sigma}$, the perturbation parameter, and $1/\min_i \|\beta\|_i$, the norm of the smallest parameter vector. This shows the polynomial dependencies on these parameters in our upper bound cannot be removed, and in particular, prove a qualitative separation between the multi-parameter setting and the single parameter setting (in which a warm start is not required). Both of our lower bounds are in the fully stochastic setting – i.e. they are based on instances in which contexts are drawn from a fixed distribution, and do not require that we make use of an adaptive adversary. First, we focus on the perturbation parameter $\sigma$.

**Theorem 6.1.** *Suppose greedy is given a warm start of size $n \leq \left( \frac{1}{100\sigma^2 \ln \frac{\rho}{100}} \right)$ in the $\sigma$-perturbed setting. Then, there exists an instance for which Greedy incurs regret $\Omega(\frac{\rho}{\sqrt{n}})$ with constant probability in its first $\rho$ rounds.*

*Remark* 1. Theorem 6.1 implies for $T < \exp(\frac{1}{\sigma})$, either

- $n = \Omega\left(\mathrm{poly}\left(\frac{1}{\sigma}\right)\right)$, or
- Greedy suffers linear regret.

The lower bound instance is simple: one-dimensional, with two arms and model parameters $\beta_1 = \beta_2 = 1$. In each round (including the warm start) the unperturbed contexts are $\mu_1 = 1$ and $\mu_2 = 1 - 1/\sqrt{n}$, and so the perturbed contexts $x_1^t$ and $x_2^t$ are drawn independently from the Gaussian distributions $\mathcal{N}(1, \sigma^2)$ and $\mathcal{N}(1 - \frac{1}{\sqrt{n}}, \sigma^2)$, for $\sigma = \sqrt{\frac{1}{100n \ln \frac{\rho}{100}}}$. We show the estimators after the warm start have additive error $\Omega\left(\frac{1}{\sqrt{n}}\right)$ with a constant probability, and when this is true, with constant probability, arm 1 will only be pulled $\tilde{O}\left(n^{2/3}\right)$ rounds. So, with constant probability greedy will pull arm 2 nearly every round, even though arm 1 will be better in a constant fraction of rounds.

We now turn our attention to showing that the warm start must also grow with $1/\min_i ||\beta_i||$. Informally, the instance we use to show this lower bound has unperturbed contexts $\mu_i^t = 1$ for both arms and all rounds, and $\beta_1 = 8\epsilon, \beta_2 = 10\epsilon$. We show again that the warm start of size $n$ yields, with constant probability, estimators with error $\frac{c_i}{\sqrt{n}}$, causing Greedy to choose arm 2 rather than arm 1 for a large number of rounds. When 2 is not pulled too many times, with constant probability its estimate remains small and continues to be passed over in favor of arm 1.

**Theorem 6.2.** *Let $\epsilon = \min_i |\beta_i|$, $\sigma < \frac{1}{\sqrt{\ln \frac{T}{\delta}}}$ and $\frac{T}{\delta} < 2^{n^{1/3}}$. Suppose Greedy is given a warm start of size $n \leq \frac{1}{2\epsilon}$. Then, some instances cause Greedy to incur regret*

$$R(T) = \Omega\left(\epsilon\left(e^{\frac{1}{18\sigma^2}} - n^{\frac{2}{3}}\right)\right).$$

*Remark* 2. Observe again that this implies that Greedy can be forced to incur linear regret if its warm start size does not grow with $1/\min_i ||\beta_i||$ for exponentially many rounds.

## Footnotes

[1] The full version of this paper is available at https://arxiv.org/abs/1801.03423.

[2]To convert a multi-parameter problem to single-parameter, concatenate the parameter vectors $\beta_i \in \mathbb{R}^d$ into a single vector $\beta \in \mathbb{R}^{kd}$, and lift contexts $x_i^t$ into $kd$ dimensions with zeros in all irrelevant $kd - d$ coordinates.

[3]Bastani et al. [2017] assume a single context at each round, shared between two actions. We consider each action as parameterized by its own context, and $k$ can be arbitrary.

[4]A random variable $Y$ with mean $\mu$ is $s^2$-subgaussian if $\mathbb{E}\left[e^{t(Y-\mu)}\right] \leq e^{t^2 s^2/2}$ for all $t$.

[5]The notation is chosen since $\mu_i$ will be the mean around which the perturbed context is drawn.

[6]E.g. for margins, consider the one-dimensional case: a lower truncated Gaussian tightly concentrates on its minimal support value.

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
