[Reviews · NeurIPS 2018]

Reviewer 1



The work studies the greedy algorithm for linear contextual bandits under smoothed analysis. The contexts are chosen adversarially, and are then perturbed by a zero-mean i.i.d Gaussian. The reward per arm are chosen stochastically with expectation that is a linear function of the context. The authors differentiate between the single parameter setting in which the same function is used for all arms, and the multiple paramter setting in which a different function is used for each arm. The behavior of the algorithm is different in each model. In the single parameter setting, exploring one arm can provide information regarding the expected reward of all other arms. In the multiple parameter setting this is not the case, and the authors show that the algorithm can lead to $\sqrt{T}$ regret only after a sufficiently long warm-start period. The Gaussian perturbations ensures two things. First, that the second moment matrix of the context vectors is well-conditioned. This allows the least square estimator to properly identify the model parameters within reasonable sample complexity. Second, that there is nonneligible probability that the reward of the optimal arm is well-separated from the rewards of the rest of the arms. In the multiple parameter model, after the warm-start, this allows the greedy algorithm to pull the optimal arm in sufficiently many rounds which promotes exploration. The paper is well-written. The results seem novel and interesting. I am missing a discussion about the motivation behind the analysis: why and in what way exactly is the greedy algorithm "fair" (simply picking an action uniformly at random can also be considered fair). Minor comments: -------------- Line 243: must rounds Throughout the paper, there seems to be some confusion regarding column and row vectors. I ask the authors to please use consistent notation.

Reviewer 2



The paper asks an important question about need for explicitly exploration of validity of attractive greedy algorithm for linear contextual multi-armed bandit settings. While it is clear that greedy algorithm can have linear regret in worst-case, it is observed to be attractive in practice. To characterize this observation theoretically, the authors use smoothed analysis framework, and show that the greedy algorithm indeed as near-optimal regret (depending inversely on variance of Gaussian perturbation, as is common in smoothed analysis). I think the result is extremely powerful. There have been some attempts to prove results in this direction, but the assumptions made here are substantially more general and elegant through use of smoothed analysis framework. Further an accompanied lower provides complete understanding of greedy algorithm in this setting. The results for multi-parameter setting are not as clean as the single parameter but still very strong. Post-rebuttal: I did not have any pressing concerns to be addressed by author feedback. After reading the author feedback, my score remains the same.

Reviewer 3



Post Rebuttal: Thanks for the clarification. You are right about the lemma. Sorry about this. I raised my score correspondingly. In my defense, it does seem like the lemma can be improved, however. Using the method of mixtures the dependence on d in the square root should disappear. Perhaps there is a small price to pay in the logarithmic term. Here is the brief sketch: Let eta_1,...,eta_n be a sequence of conditionally 1-subgaussian random variables adapted to filtration (F_t). Then let X_1,...,X_n be a sequence of random elements with X_t in R^d and assume that X_t is F_{t-1}-measurable (that is, (X_t) is (F_t)-predictable). Let us fix 1 <= i <= d. Since X_{ti} is F_{t-1}-measurable and eta_t is conditionally 1-subgaussian it follows that X_{ti} eta_t is conditionally X_{ti}-subgaussian. Hence exp(sum_t (L X_{ti} eta_t - L^2 X_{ti}^2 / 2)) is a supermartingale for any L > 0. Hence Prob(sum_t X_{ti} eta_t >= L sum_t X_{ti}^2 / 2 + log(1/delta) / L) <= delta for any L > 0. If the X_{ti} where non-random, then you could choose L \approx sqrt(log(1/delta) / sum_t X_{ti}^2). But you can't do this, because it is random. The solution is either the method of mixtures or a covering argument. The first is cleaner (Google it). The covering argument is easier to explain simply. Since 0 <= sum_t X_{ti}^2 <= n we can take a union bound over all L in some covering set of [0,n] shows that Prob(sum_t X_{ti} eta_t >= something_1 + sqrt(sum_t X_{ti}^2 log(something_2 /delta)) where something_2 should be a lower degree polynomial of n and something_1 is small due to the approximation error in the covering set. Now take a union bound over the coordinates and argue symmetrically for the lower tails and you should have something like |sum_t X_{ti} eta_t| lesssim sqrt(sum_t X_{ti}^2 log(d n / delta)) Hence || sum_t eta_t X_t ||_2^2 <= sum_i sum_t X_{ti}^2 log(dn / delta) = n log(dn/delta). And so you should have a bound that is O(sqrt(n log(dn / delta))). I did not go back to check how much this would improve the results. I also encourage you to work hard on motivating the assumptions and correcting the minor notational issues. Summary. This paper is about contextual linear bandits. The authors are trying to understand the phenomenon observed in practice that exploration seems much less important than the theory usually suggests. To make a start understanding this they analyze the greedy policy that chooses the arm for which the inner product with the least squares estimator and the context is largest. With no further assumptions this leads to linear regret. The authors make a 'perturbation' assumption where the contexts are first chosen by an adversary and then perturbed using zero-mean noise. Under carefully chosen assumptions on the noise they show that the greedy algorithm now enjoys O(sqrt(T)) regret. The two standard settings are studied. (a) where there is one unknown parameter vector and (b) where there is an unknown parameter vector for each action. The results setting (b) are unsurprisingly much weaker than (a) where some forced exploration of all arms is required. A lower bound is given proving that this is unavoidable for the greedy algorithm. In setting (a) the assumptions on the perturbation distribution ensure the minimum eigenvalue of the design matrix is increasing linearly, which ensures sufficient accuracy of the least-squares estimator. Quality: The results seem reasonable in principle. I think there is at least one bug in the proof (see 'Question about Lemma A1' below). It should be fixable with limited loss in the logarithms. Originality: Perturbation analysis of this kind is not new, but I think the modifications are interesting enough. Clarity: This is a weak point of the paper. There are issues with row/column vectors being mismatched, notation used in multiple different ways. Many unexplained details. Regret bounds and conditions under which they hold are given in Big-O notation and there is no way to see what is the variable of interest. The supplementary material is in better shape than the main body, which is a bit unfortunate. Statements involving probabilities are often a little imprecise and there are many typos. The full paper is also very long. I have to say I did not read all proofs. Significance: The difference between practice and theory for contextual bandits is quite large and any attempt to understand this is welcome and potentially significant. For the present article the main issue is the validity of the assumptions. Is the perturbation model reasonable? I did not see that this is addressed anywhere, which is quite unfortunate. Other comments: I leave the philosophy to others, but is it not also repugnant and short-sighted to explore suboptimally in clinical trials and then waste the results because the data does not statistically support the hypothesis that the drug works? I thought the point of sequential design was to optimize this process. Perhaps a better argument is that explicit exploration may actually not be required. That is, explicit exploration is not the enemy, but rather excessive exploration. Question about Lemma A.1: The proof of this result does not seem correct. You are right that the coordinates of S_t are rs-subgaussian, but this is much weaker than you need because summing over the squares will give you a bound of sqrt(2trsd log(d/delta)), which has an extra d relative to what you have. Really what you want to show is that with high probability sum_{s=1}^t eta_s x^s_i < sqrt(sum_{s=1}^t eta_s (x^s_i)^2 log(1/delta)) Then summing the squares of the coordinates and taking the sqrt will yield your bound. The problem now is that showing the above for arbitrary choices of x the above might not hold. You will need a more sophisticated method such as the method of mixtures and something extra will appear in the logarithm. Overall I find the paper to be interesting, but suffers from several issues in terms of clarity and motivation for the assumptions. Having found so many issues in the first 20 pages or so it seems the authors should spend more time polishing the article and perhaps consider submitting to a journal where a work of this length might be more appreciated. Minors: * Is mathcal H^* defined? * Supp. First para of proof of Lemma 3.2 there is a missing \Vert * There seems to be some indecision about what is a column/row vector. Eg., L215 there is ^beta^t x_i^t * Display after L185. beta e -> beta . e. * Should say that s-subgaussian means exp(lambda X) <= exp(lambda^2 s / 2) not exp(lambda^2 s^2 / 2)? Unfortunately different authors use different defs (I see it in the supp, but not main) * Sometimes you write || . || and sometimes || . ||_2 for the Euclidean norm. * Condition 1 (and elsewhere). You can help the reader by giving domains for variables. * The wide hats over r-good and r-auspicious don't seem to add anything and look bad. * Theorem 4.1. The definition of low perturbation regime is given in terms of Big-O notation and the regret here is also Big-O notation. But what is the variable? T? d? The theorem statement should be made rigorous. * Expectation operators are often used when the underlying measure is unclear and sometimes the notation is really strange (Proof of Lem 3.2 for example). * Second to last equality in Proof of Lem 3.2. Why is this an equality? Diversity provides a lower bound, no? * Why is there a t in logarithm in Lemma A.1? The union bound is done coordinate-wise, not over time so it should be log(d/delta). * Lemma A.2. This proof is missing some expectations. The LHS is a number, but the RHS after the second inequality is a random variables. * Proof of Lemma 3.1. Is eta^t defined? I suppose it must be the vector of noise up to time t. Note that this lemma probably needs changing if Lemma A.1 is indeed not true. Elsewhere it seems that eta^t is the vector of noise over the coordinates, so this seems like a bit of a mismatch. * The statement of Corollay 3.1 is a bit weird. Whether or not a certain number of rounds are r-auspicious is itself a random quantity, so the probabalistic quantifier should not come after. Perhaps you need to define an indicator function IND that this number of rounds occurs and then claim that with probability at least 1 - delta it holds that IND ||beta - ^beta^t|| \leq 32 \sqrt .... * L223. The probability measure here should be given. This is over the perturbations of the adversary in round t only.